# RILP Induces Cholesterol Accumulation in Lysosomes by Inhibiting Endoplasmic Reticulum–Endolysosome Interactions

**DOI:** 10.3390/cells13161313

**Published:** 2024-08-06

**Authors:** Yang Han, Xiaoqing Liu, Liju Xu, Ziheng Wei, Yueting Gu, Yandan Ren, Wenyi Hua, Yongtao Zhang, Xiaoxi Liu, Cong Jiang, Ruijuan Zhuang, Wanjin Hong, Tuanlao Wang

**Affiliations:** 1State Key Laboratory of Cellular Stress Biology, School of Pharmaceutical Sciences, Fujian Provincial Key Laboratory of Innovative Drug Target Research, Xiamen University, Xiamen 361102, China; 18900960635@163.com (Y.H.); 18862083375@163.com (X.L.); 13870041396@163.com (L.X.); wzh1002028777@163.com (Z.W.); guyueting09@163.com (Y.G.); renyd123@163.com (Y.R.); huawenyi1999@163.com (W.H.); z82425995@163.com (Y.Z.); 32320221154469@stu.xmu.edu.cn (X.L.); jiangcong05062000@163.com (C.J.); rjzhuang0715@163.com (R.Z.); 2Institute of Molecular and Cell Biology, A*STAR (Agency of Science, Technology and Research), Singapore 138673, Singapore

**Keywords:** RILP, organelle interaction, cholesterol transport, autophagy, Rab7

## Abstract

Endoplasmic reticulum (ER)–endolysosome interactions regulate cholesterol exchange between the ER and the endolysosome. ER–endolysosome membrane contact sites mediate the ER–endolysosome interaction. VAP-ORP1L (vesicle-associated membrane protein-associated protein- OSBP-related protein 1L) interaction forms the major contact site between the ER and the lysosome, which is regulated by Rab7. RILP (Rab7-interacting lysosomal protein) is the downstream effector of Rab7, but its role in the organelle interaction between the ER and the lysosome is not clear. In this study, we found RILP interacts with ORP1L to competitively inhibit the formation of the VAP–ORP1L contact site. Immunofluorescence microscopy revealed that RILP induces late endosome/lysosome clustering, which reduces the contact of endolysosomes with the ER, interfering with the ER–endolysosome interaction. Further examination demonstrated that over-expression of RILP results in the accumulation of cholesterol in the clustered endolysosomes, which triggers cellular autophagy depending on RILP. Our results suggest that RILP interferes with the ER–endolysosome interaction to inhibit cholesterol flow from the endolysosome to the ER, which feedbacks to trigger autophagy.

## 1. Introduction

In eukaryotic cells, the endomembrane system is highly partitioned into morphologically and functionally distinct membrane-bound organelles. These organelles are inter-connected through organelle–organelle interactions, which are mediated by membrane contact sites (MCSs) [1]. The endoplasmic reticulum (ER) spreads throughout the cytoplasm from the perinuclear space to the plasma membrane to form a cellular ER network, which interacts with many other organelles (i.e., the mitochondria, the endosome, the lysosome, the peroxisome, and the plasma membrane) [2,3]; therefore, the ER plays predominant roles in organelle–organelle interactions.

Membrane contact sites are tethers for organelle–organelle interactions, but not for membrane fusion, which is constructed by protein–protein interactions or a protein interaction complex [4]. The ER–endolysosomal membrane contact sites are established by membrane receptors from the ER and proteins from the endosomes/lysosomes. The major sperm protein (MSP) domain-containing family, vesicle-associated membrane protein-associated proteins A and B (VAPa and VAPb), and motile sperm domain-containing protein 2 (MOSPD2) interact with FFAT motif-containing proteins, such as Stard3, ORP1L (OSBP-related protein 1L), and OSBP, respectively, which play major roles in the formation of ER–endolysosomal membrane contact sites [5,6,7,8,9,10]. Other ER proteins, such as protrudin, TMCC1 (transmembrane and coiled-coil protein 1), PTP1B (protein tyrosine phosphatase 1B), VPS13 (vacuolar protein sorting 13), RNF26 (ring finger protein 26), PDZ8 (PDZ domain containing 8), and IP3R (inositol1,4,5-triphosphate receptor), were identified to build connections with the endosome/lysosome [11,12,13,14,15].

ER–endolysosome contact (interaction) regulates multiple cellular events (i.e., endosomal trafficking, lipid transfer, and cellular signaling). ER-resident protein TMCC1 interacts with endosomal CORO1C (coronin 1c) to drive endosome tubulation and fission [16,17]. PTP1B targeting of EGFR modulates EGFR (epidermal growth factor receptor) signaling [18]. IP3R serves as a calcium channel to mediate calcium transfer between two organelles [19]. Lipid transfer across ER–endolysosome contacts is probably the main function of ER–endolysosome interactions. VAP proteins interact with OSBP or ORP1L to regulate cholesterol transfer from the endosome/lysosome to the ER [20,21]. VAP proteins also interact with Stard3 to mediate cholesterol transport from the ER to the endosome [22,23]. Protrudin–Rab7 contact sites may be engaged in phospholipid transfer [13,24].

Although ER–endolysosomal membrane contact sites are constructed by one ER-located protein and another protein from the endolysosome, other accessory proteins are characterized to be essential in supporting the formation of the MCSs. For VAP–ORP1L contact sites, Rab7 regulates the recruitment of ORP1L to the cytoplasmic side of the late endosome/lysosome [25]. RILP (Rab7-interacting lysosomal protein), the most important downstream effector of Rab7, regulates the biogenesis of the late endosome/lysosome and lysosomal positioning [26]. RILP also binds to the dynein/dynactin motor protein to drive minus-end directed transport of the late endosome/lysosome [27]. However, the role of RILP in ER–endolysosome interactions remains unclear.

In this study, we found that RILP interacts with ORP1L to competitively inhibit the formation of the VAP–ORP1L contact site. RILP induces late endosome/lysosome clustering, which reduces the contact of endolysosomes with the ER. Furthermore, overexpression of RILP results in the accumulation of cholesterol in the clustered endolysosomes, which triggers cellular autophagy.

## 2. Materials and Methods

### 2.1. Antibodies and Reagents

Rabbit polyclonal antibody against RILP used for Western blotting was generated by GenScript (Nanjing, China). The RILP antibody in the immunoelectron microscopy assay was purchased from Sigma-Aldrich (St. Louis, MO, USA, cat.no. SAB3500660). Rabbit polyclonal antibody against Rab7 (Proteintech, Wuhan, China, cat.no. 55469-1-AP), monoclonal antibodies (mAb) against GAPDH (60004-1-Ig, G1100, LABLEAD lnc. Beijing, China), mAb against α-tubulin (cat.no. 66031-1-Ig), GFP (cat.no. 66002-1-Ig), Rabbit polyclonal antibody against mCherry (cat.no. 26765-1-AP), mAb against His-tag (cat.no. 66005-1-Ig), Rabbit polyclonal antibody against VAP-A (cat.no. 15275-1-AP), and Rabbit polyclonal antibody for p62 (cat.no. 18420-1-AP) were purchased from Proteintech (Wuhan, China). Rabbit polyclonal antibody against LAMP1 was from ABclonal (Wuhan, China, cat.no. A21481). Rabbit monoclonal antibody against KDEL was purchased from Abcam (Boston, UK, ab176333). mAb against myc tag (9E10) was obtained from ATCC (Manassas, VA, USA). HRP-conjugated, Cy5-conjugated, and Texas red-conjugated secondary antibodies were from Jackson Immuno Research (cat. 111-035-003, 115-025-003, 111-175-144, 115-295-003, West Grove, PA, USA). Rabbit polyclonal antibodies for LC3 (cat.14600-1-AP) and mAb for LC3 (M152-3) were from Abcam (Boston, UK). mTOR Rabbit pAb (A2445), Phospho-mTOR-S2448 Rabbit mAb (AP0115), ULK1 Rabbit pAb (A8529), and Phospho-ULK1-S757 Rabbit pAb (AP0736) were purchased from ABclonal (Wuhan, China).

GST-Sepharose 4B resin was from GE Healthcare (AL, USA, cat.no. 45-000-139). Filipin complex, cholesterol, U18666A, MG132, and chloroquine (CQ) were purchased from Sigma-Aldrich. Torin1 was purchased from Targetmol (Boston, MA, USA, cat.no. T6045). All oligonucleotides are listed in the Appendix A.

### 2.2. Expression Plasmids

GFP-RILP, GFP-RILP (1-198aa, N-terminal, NT), GFP-RILP (199-401aa, C-terminal, CT), mCherry-RILP-NT, and mCherry-RILP-CT were described previously [28]. mCherry-RILP and myc-RILP were constructed by subcloning RILP cDNA into the pmCherry vector and pDmyc vector, respectively. VAPa, ORP1L, ORP7, or their truncated mutants were constructed by subcloning the correspondent coding region into a pEGFP-C1 vector to generate GFP-tagged protein expression plasmids, respectively. His-tagged protein expression plasmids were constructed using a similar method into pET-28a plasmids. mCherry-VAP-A was constructed by subcloning VAP-A cDNA into the pmCherry vector. GST-tagged VAP-A(1-225aa) and GST-RILP plasmids were generated by subcloning the correspondent coding region into pGEX-4T-1 vectors, respectively. The coding region for Sec61β was retrieved from cDNA derived from HEK-293T cells and subcloned into pmCherry-C1 vectors to generate mCherry-sec61β expression plasmids. The mCherry-LC3 plasmid is from Dr. Wanjin Hong (Institute of Molecular and Cell Biology, Singapore). All constructed plasmids were confirmed by DNA sequencing.

### 2.3. Cell Culture and Transfection

Hela and HEK-293T cell lines were from ATCC (American Type Culture Collection). Cells were cultured in Dulbecco’s Modified Eagle’s Medium (DMEM) supplied with 10% fetal bovine serum in a humidified incubator with 5% CO_2_ and maintained at 37 °C. Cells were transfected with the indicated plasmids by using HighGene plus Transfection reagent (ABclonal) according to the manufacturer’s instructions. For the inhibitor treatments, Hela cells were treated with CQ (20 µM) or MG132 (20 µM) for the indicated time under normal culture conditions. For cholesterol-related experiments, cells were treated with cholesterol (10 mg/mL) or U18666A (3.0 μg/mL). For cholesterol-depleting conditions, Hela cells were cultured in DMEM supplemented with 5% lipoprotein-deficient serum, 230 μM mevalonate, and 50 μM lovastatin to inhibit HMG-CoA (3-hydroxy-3-methyl-glutaryl-CoA) reductase.

### 2.4. Adenoviral-Mediated Gene Expression System and pSicoR/pCDH-Mediated Gene Lentivirus Expression System

The adenovirus was prepared using the AdEasy system for generating recombinant adenovirus [29]. Briefly, the coding regions of RILP and Rab7 were cloned into the pAdTrack- cytomegalovirus vector and then linearized. The linearized plasmids were transformed into the competent AdEasier *E. coli* cells to generate recombinant adenovirus plasmids. The recombinant adenovirus plasmids were transfected into 293A cells to produce recombinant adenovirus (referred to as Ad-RILP and Ad-Rab7 below). For adenovirus-mediated gene knockdown, shRNA-RILP and shRNA-Rab7 were cloned into pAdTrack-H1-U6 vectors, respectively. The shRNA sequences are listed in the Appendix A.

For lentivirus-mediated protein expression, the overexpression of RILP was achieved using the pCDH-CMV-MCS-EF1-Puro vector-mediated lentivirus expression system [30]. mCherry-RILP was cut out from the mCherry-C1 vector with NheI/BamH1 and subcloned in the same enzyme sites of the pCDH-CMV-MCS-EF1-Puro vector. For virus preparation, HEK-293T cells were transfected with a lentiviral backbone and helper plasmids (pMD2.G and psPAX2) for 48 h and then the culture media was collected. Inoculate cells on a 6-well plate and add 1ml lentivirus stock solution after stable growth. After 24 h, replace with fresh culture medium. After 72 h of transfection, the supernatant is collected and filtered with a 0.45 µm filter (Millipore) to obtain the concentrate. Western blotting was used to verify the expression level of the target protein.

shRNA targeting sequences were used for gene knockdown of RILP or Rab7 through a lentiviral vector pSicoR-mediated gene knockdown system, respectively [31]. The preparation method for the virus is the same as the overexpression system mediated by the pCDH-CMV-MCS-EF1-Puro vector-mediated lentivirus expression system mentioned above.

### 2.5. GST-Pulldown and Western Blot

The GST-pulldown assay is generally used to examine protein interactions. HEK-293T cells were transfected with the indicated plasmids and then lysed with lysis buffer (containing 20 mM HEPES, pH 7.4, 1% Triton X-100, 100 mM NaCl, 5 mM MgCl_2_, and EDTA-free proteinase inhibitor cocktail) for 1 h on ice. The cell lysates were centrifuged at 13,000× *g* for 30 min at 4 °C. The supernatants were incubated with GST-fusion protein coupled to GST-Sepharose 4B resin at 4 °C overnight. GST-Sepharose 4B resin was washed three times using the lysis buffer mentioned above containing different concentrations of NaCl (500, 300, and 100 mM). The bound proteins were analyzed using a Western blot assay. The GST fusion protein was stained with Coomassie Brilliant Blue (Sigma Aldrich, USA).

For the in vitro binding assay, the prokaryotic expressed His-RILP and His-ORP1L/truncated mutants were purified and then incubated with the GST-VAPa(1-225aa) retained on the GST beads. The amount of ORP1L bound to VAPa was assessed by Western blot.

For Western blot assays, proteins were resolved by SDS-PAGE and transferred onto a PVDF membrane. Membranes were blocked with 5% milk and incubated with specific primary antibodies, followed by peroxidase-conjugated secondary antibodies. The blots were visualized by an ECL kit (Pierce, Rockford, IL, USA). Subsequently, grayscale analysis for the protein bands was conducted using Gel-Pro Analyzer Version 6.3 Gel-Pro software. 

### 2.6. Transmission Electron Microscopy (TEM) and Immunoelectron Microscopy (IEM)

Hela cells infected with Ad-vector or Ad-RILP for 6 h or lentivirus expressing pCDH-cherry-RILP were processed for transmission electron microscopy (TEM) analysis. The ultrathin-sectioned samples were analyzed using a transmission electron microscope (HT7800 RuliTEM) as described [26].

The immunogold staining experiment was carried out as described [32]. Briefly, Hela cells were fixed, embedded, and sectioned, then the sections were labeled with the primary antibody followed by the secondary antibody (Nanogold^®^-IgG goat anti-rabbit IgG (H+L), Nanoprobes). The immuno-labeled sections were ultrathin-sectioned and processed by a transmission electron microscope.

### 2.7. Immunofluorescence Microscopy

Immunofluorescence microscopy was performed as previously described [33]. Briefly, cells grown on coverslips were washed with PBSCM (PBS containing 1.0 mM CaCl_2_ and 1.0 mM MgCl_2_) 3 times and then fixed with 4% paraformaldehyde (PFA) for 20 min, before being neutralized with NH_4_Cl and permeabilized with 0.1% Triton-X 100 (Sigma, St. Louis, MO, USA). Then, cells were incubated with primary antibodies diluted in FDB (BSCM containing 5% goat serum, 5% fetal bovine serum, and 2% BSAP), followed by fluorophore-conjugated secondary antibodies at room temperature for 1 h in the dark. The immunolabeled cells were analyzed with a Zeiss LSM980 laser confocal microscope (Carl Zeiss, Germany).

For Filipin staining, Hela cells were fixed with 4% PFA, rinsed with glycine, and then incubated with 50 μg/mL Filipin (dissolved in dimethyl sulfoxide and diluted with FDB to a working concentration of 50 μg/mL) to detect free cholesterol, avoiding bright light [34,35]. A Zeiss LSM980 laser confocal microscope and a Leica fluorescence microscope were used for image acquisition.

### 2.8. High Intelligent and Sensitive Structured Illumination Microscope (HIS-SIM)

Cells on cover glasses at 70–80% confluence were transfected with mCherry-sec61β, then infected with Ad-vector, Ad-RILP, Ad-shRNA-Ctrl, or Ad-shRNA-RILP [36]. Cells were fixed with pre-cooled methanol and then stained with Lamp1 antibody. The samples were analyzed by an HIS-SIM instrument. ImageJ_v1.8.0 and Imaris software (X64 9.2.0) were used to analyze the endoplasmic reticulum–endolysosome membrane contacts.

### 2.9. Statistical Analysis

Quantitative data were presented as means ± standard deviations (SDs). Statistical significance was estimated by a two-tailed Student’s *t*-test and analysis of variance (ANOVA). The mean values of the two groups were considered significantly different at * *p* < 0.05 or ** *p* < 0.01.

## 3. Results

### 3.1. RILP Interacts with ORP1L to Interfere with Its Interaction with VAP

Both RILP and ORP1L are important downstream effectors of Rab7. Rab7 recruits RILP and ORP1L at the late endosome/lysosome membrane to form the Rab7–RILP–ORP1L complex. Rab7 is crucial for the formation of the VAP–ORP1L ER–endolysosomal membrane contact site; however, the action of RILP on VAP–ORP1L interactions remains unclear. We examined whether RILP binds to ORP1L. Cell lysates containing GFP-ORP1L were subjected to a GST-pulldown assay using GST-tagged RILP (Figure 1A) or its truncated forms (Figure 1B). The results demonstrated that ORP1L efficiently binds to RILP and Rab7 (Figure 1A). In addition, ORP1L binds preferentially to the amino-terminal region of RILP (Figure 1B). To identify which region of ORP1L is responsible for interaction with RILP, the truncated forms GFP-ORP1L(1-468aa), GFP-ORP1L(1-485aa), and GFP-ORP1L(486-950aa) were constructed and processed for the GST-pulldown assay. As shown in Figure 1C, the full-length ORP1L and ORP1L(1-485aa), but not ORP1L(1-468aa) and ORP1L(486-950aa), can efficiently interact with RILP. Because ORP1L(1-468aa) and ORP1L(486-950aa) lack the FFAT (two phenylalanines in an acidic tract) motif, the results suggest that RILP interacts with ORP1L via the FFAT motif.

Immunofluorescence microscopy revealed that ORP1L normally disperses in the cytoplasm, with an equivalent pool of ORP1L co-localizing with Lamp1 (endolysosomal marker) (Appendix A). However, mCherry-RILP co-localized with GFP-ORP1L and resulted in ORP1L clustering at the perinuclear region in Hela cells (Figure 1D). Furthermore, RILP(1-198aa) co-localized with ORP1L much more than RILP(199-401aa) (Figure 1D), which is consistent with the observation that the N-terminal region of RILP is preferentially involved in the interaction with ORP1L. As expected, RILP co-localizes with ORP1L(1-485aa), but not ORP1L(1-468aa) or ORP1L(486-950aa) (Figure 1E), which is consistent with the interaction results. These results are consistent with the observation that RILP interacts with the full length of ORP1L and the truncated ORP1L(1-485aa) mutant, both containing the FFAT motif.

As ER protein VAP interacts with ORP1L through binding to the FFAT motif, the interaction of RILP with ORP1L (via the FFAT-dependent pathway) probably influences the VAP–ORP1L interaction. To test this hypothesis, GFP-ORP1L was co-transfected with increasing amounts of myc-RILP in Hela cells, and the cell lysates were processed for GST-pulldown assays using GST-VAPa(1-225aa) (the cytoplasmic region). The results demonstrated that the amount of ORP1L binding to VAPa decreased with the increasing amount of RILP co-expressed (Figure 2A,B), indicating that RILP inhibits ORP1L binding to VAPa likely via competing with binding to the FFAT motif in a mutually exclusive manner. To further confirm this result, recombinant GST-VAPa was used to bind His-ORP1L or the truncated ORP1L(336-950aa) in the presence of increasing amounts of His-RILP in vitro. The results again confirmed that RILP influences VAPa binding to either ORP1L or ORP1L(336-950aa) (Figure 2C,D). These results suggest that RILP may negatively regulate the ER–endolysosome interaction by suppressing the formation of the VAP–ORP1L contact site.

In addition, imunofluorescence microscopy showed that ORP1L co-localizes with VAPa (Figure 2E, upper panels), but RILP does not co-localize with VAPa or affect the distribution of VAPa (Figure 2E, middle panels). Intriguingly, overexpression of RILP decreased ORP1L co-localization with VAPa (Figure 2E, lower panels), which is consistent with the above results that RILP interferes with the VAP–ORP1L interaction.

### 3.2. RILP Interferes with ER–Late Endosome/Lysosome Contact

The ER network spreads throughout the cytoplasm from the perinuclear space to the plasma membrane. Morphologically, the lamellar ER sheets surround the perinuclear region, and the tubular ER spreads out and connects with the plasma membrane. It is evident that ER–organelle interactions are usually observed between the peripheral tubular ER and the peripheral endo/lysosome [37]. To examine the effects of RILP on ER–late endosome/lysosome contact, Hela cells were infected with PCDH-CMV-Cherry-RILP to ensure that all cells expressed RILP (Appendix A). Then, cells were transfected with mCherry-Sec61β to reveal the ER network. The Lamp1 antibody was used to label the late endosomes/lysosomes. We applied high intelligent and sensitive structured illumination microscopy (HIS-SIM) to view the ER network and ER–late endosome/lysosome contact. A total of 30 cells were observed, and the contact sites were quantified. Immunofluorescence microscopy revealed that the peripheral late endosomes and lysosomes tend to come into contact with ER tubules in Hela cells (Figure 3A). However, it is observed that the majority of late endosomes/lysosomes are distributed at the perinuclear region, and the amount of peripheral late endosomes/lysosomes are greatly reduced upon overexpression of RILP in Hela cells, indicating ER–endolysosomal membrane contacts are significantly decreased (Figure 3A). This is consistent with our early studies showing that RILP regulates lysosomal positioning in the cell [38]. On the contrary, when the expression of RILP was inhibited by shRNA-RILP (Appendix A), the number of late endosomes/lysosomes in the perinuclear region reduced, and the number of peripheral late endosomes/lysosomes increased, suggesting that depletion of RILP increases ER–endolysosomal membrane contact (Figure 3B). These results suggest that RILP inhibits ER–late endosome/lysosome interactions via molecularly interfering with the VAP–ORP1L interaction.

Interestingly, while the expression of Rab7 was knocked down by lentivirus-mediated expression of shRNA-Rab7 (Appendix A), RILP still results in clustering of the late endosomes/lysosomes (Appendix A), and consequently decreases ER–endolysosome contact (Figure 3C), suggesting that RILP may interfere with ER–endolysosome interactions independently of the activity of Rab7 or likely through interacting with other Rab GTPases such as Rab34 or Rab36.

### 3.3. RILP Induces Cholesterol Accumulation in the Late Endosomes/Lysosomes

ER–endolysosome interactions regulate cholesterol transfer from the late endosome/lysosome to the ER [21]. Therefore, inhibition of the formation of the ORP1L–VAPa contact site by RILP likely influences cholesterol transport. Immunofluorescence microscopy revealed that overexpression of GFP-RILP induces cholesterol (referred to as free cholesterol labeled by Filipin) accumulation and co-localization with RILP in Hela cells (Figure 4A). Further examination showed that cholesterol is accumulated in the late endosomes/lysosomes marked by Lamp1 (Figure 4B). Interestingly, RILP(1-198aa), but not RILP (199-401aa), has the same effects as the wildtype of RILP on the distribution of cholesterol (Figure 4C), consistent with the notion that RILP regulates cholesterol transfer via interfering with ER–lysosome contact via suppressing VAP–ORP1L interaction. RILP knockdown resulted in a more peripheral distribution of cholesterol (Appendix A), which is consistent with RILP knockdown increasing the number of peripheral late endosomes/lysosomes and facilitating ER–endolysosome contact. Additionally, overexpression of RILP can still induce cholesterol accumulation in the lysosome upon Rab7 depletion by siRNA-Rab7 (Figure 4D), indicating that RILP induces cholesterol accumulation in the late endosomes/lysosomes in a Rab7-independent manner. Together, these data suggest that RILP inhibits cholesterol export from the late endosomes/lysosomes to the ER through interfering with ER–endolysosome interactions.

### 3.4. Accumulation of Cholesterol Inhibits ER–Endolysosomal Contact

We next examined contact of the late endosomes/lysosomes with the ER under low- or high-cholesterol conditions. When Hela cells are grown in low-cholesterol media, the late endosomes/lysosomes are more dispersed throughout the cytoplasm, which usually comes into contact with the ER (Figure 5A). However, high cholesterol levels in the culture induce the perinuclear distribution of the late endosomes/lysosomes marked by Lamp1 (Appendix A), which significantly decreased contact of the late endosomes/lysosomes with the ER (Marked by Sec61β) (Figure 5B), suggesting that a high-cholesterol condition inhibits ER–endolysosome interactions. Next, we tested whether inhibition of cholesterol transport by U18666A treatment disrupts ER–endolysosomal contact. U18666A is a drug that inhibits cholesterol trafficking, resulting in cholesterol accumulation in the late endosomes/lysosomes. As shown in Figure 5C, U18666A treatment significantly decreased ER–late endosome/lysosome contact, suggesting that high levels of cholesterol or the accumulation of cholesterol inhibit ER–endolysosome interactions.

As high cholesterol or accumulation of cholesterol inhibits the ER–endolysosome interaction, similar to the effects of RILP overexpression, there exists a possibility that high cholesterol elevates the expression of RILP, which may facilitate the clustering of late endosomes/lysosomes to the perinuclear region. To test this hypothesis, we examined the expression of RILP in Hela cells under different nutrient conditions. As shown in Figure 5D,E, the expression level of RILP is significantly elevated under high cholesterol conditions. Upon U18666A treatment, the expression level of RILP is also increased in a dose-dependent manner (Figure 5F,G). The protein level of RILP is also increased under starvation (Figure 5D) and upon Torin1 treatment (which induces autophagy) (Appendix A). In addition, the transcription of RILP is upregulated in high fat-fed type 2 diabetic mice (Appendix A). These results suggest that the expression of RILP is physiologically and dynamically regulated by cholesterol. Taken together, high levels of cholesterol may disrupt ER–endolysosomal interactions by upregulating RILP, but on the other hand, the expression of RILP may feedback to enhance cholesterol accumulation in the late endosomes/lysosomes.

### 3.5. RILP-Induced Accumulation of Cholesterol Induces Autophagy

RILP inhibits cholesterol export from the late endosomes/lysosomes by suppressing VAPa-ORP1L-mediated ER–endolysosome contact, resulting in the accumulation of cholesterol in the late endosomes/lysosomes, which may act as cellular signals to trigger cellular stress responses. Overexpression of RILP increases the expression of LC3II (a marker for autophagy) under normal conditions or chloroquine/MG132 treatments (Figure 6A,B). Immunofluorescence microscopy also revealed that GFP-RILP induces the clustering of LC3-containing compartments and LC3 co-localizing with RILP in Hela cells (Figure 6C).

Consistently, high cholesterol also significantly upregulates the expression of LC3II compared to low-cholesterol treatment (Figure 6D,E). High-cholesterol treatment or U18666A inhibits the activity of mTOR (Figure 6F–H), a regulator of autophagy. High cholesterol treatment or U18666A also inhibits the ULK1 signaling pathway (Figure 6I,J). Additionally, high cholesterol increases the LC3-containing particles (autophagosomes) in Hela cells (Figure 6K), indicating that high cholesterol or blocking cholesterol trafficking promotes autophagy. Consistently, overexpression of RILP inhibits the phosphorylation of mTOR and ULK1 (Appendix A).

Transmission electronic microscopy showed that RILP induces clustering of vacuoles in Hela cells upon lentivirus-mediated expression of RILP (which results in all cells expressing RILP) (Figure 6L, with more pictures in Appendix A–H), which is consistent with the results of immunofluorescence microscopy (Figure 6C). In addition, immuno-electronic microscopy indicates that RILP is associated with autophagosomes (marked by a double-layer membrane) (Figure 6L), which is consistent with the results that RILP induces autophagy. Consistently, RILP induces cholesterol (marked by Filipin) accumulation and co-localization with autophagosomes (marked by LC3) (Figure 6M). These results indicate that RILP inhibits ER–endolysosome interactions to block cholesterol export from the late endosomes/lysosomes, consequently triggering autophagy.

### 3.6. RILP Interacts with the ORP Family to REGULATE Cholesterol Trafficking

Through GST-pulldown assays, we systematically examined the interactions between RILP and ORP family members. RILP may interact with several other ORP members in addition to ORP1L (Appendix A). ORP7 is one of the representative ORPs with a FFAT motif, and its function remains unclear. We therefore investigated the interaction between RILP and ORP7. GST-pulldown experiments demonstrated that ORP7 interacts with RILP, as well as VAPa (Figure 7A). In addition, the truncated mutant ORP7(1-409aa) can bind to RILP; however, ORP7(1-401aa) or ORP7(410-842aa), which do not contain FFAT motifs, do not exhibit binding ability to RILP (Figure 7B), indicating that ORP7 interacts with RILP through the FFAT motif in a similar way to ORP1L.

Immunofluorescence microscopy revealed that ORP7 co-localizes with RILP and VAPa (Figure 7C). Furthermore, RILP co-localizes with ORP7(1-409aa), but not ORP7(1-401aa) or ORP7(410-842aa) (Figure 7D), which is consistent with the interaction data from GST-pulldowns.

Since ORP7 binds to cholesterol, RILP–ORP7 interactions may potentially influence cholesterol trafficking. Immunofluorescence microscopy showed that ORP7 co-localizes with filipin-marked cholesterol, and overexpression of RILP induces ORP7 or ORP7(1-409aa) and cholesterol clustering (Figure 7E). This result suggests that RILP may regulate cholesterol trafficking through interaction with other ORP proteins such as ORP7 in a similar mechanism to ORP1L.

## 4. Discussion

RILP (Rab7-interacting lysosomal protein) interacts with several Rab proteins such as Rab7, Rab34, Rab36, Rab24, and Rab26 along the endocytic pathway, serving as a common effector for these Rab GTPases. RILP also interacts with the HOPs complex and dynein/dynactin to regulate late endosomal/lysosomal trafficking and the morphogenesis of the late endocytic compartments [26,39,40,41,42]. In this study, we provided molecular, cellular, and functional evidence that RILP can bind to ORP1L to competitively inhibit ORP1L–VAP interactions, suppressing ER–endolysosome interactions.

Other members of the OSBP-related protein (ORP) family with FFAT motifs (i.e., OSBP, ORP3, ORP4, ORP6, ORP7, and ORP9) interact with VAP proteins [43,44,45,46,47], which mediates the formation of membrane contact sites between the ER and other membrane compartments such as the plasma membrane, mitochondria, and lipid droplets [48,49,50,51]. As RILP binds to ORP1L in a FFAT motif-dependent manner, RILP may potentially influence other membrane contacts through mediating other VAP–ORP interactions via binding to the FFAT motif in other members. In our experiments, RILP binds to ORP3 and ORP7; however, the functional regulation of these other members by RILP needs further investigation.

Perinuclear ER sheets are usually anchored by ribosomes, which are not suitable for interaction with other organelles, including endosomes/lysosomes. On the other hand, the peripheral tubular ER usually comes into contact with other organelles (i.e., the mitochondria, the peroxisome, lipid droplets, the plasma membrane, the endosome, and lysosomes) [37,52]. Based on this understanding, we examined ER–endolysosomal contact. RILP induces clustering of the late endosomes and lysosomes, resulting in the reduction of the number of peripheral endosomes/lysosomes, therefore decreasing ER–endolysosomal contact.

Rab7 interacts with RILP and ORP1L at the late endosomes/lysosomes to form a tripartite complex, and RILP binds to the dynein/dynactin subunit, recruiting dynein motor to drive minus end-directed movement of the late endosomes/lysosomes [26,27]. Under lower cholesterol conditions, ORP1L interacts with VAP to remove the dynein motor and promotes ER–endolysosome interactions [53]. It seems that the interaction of Rab7-RILP-ORP1L with the dynein motor is sufficient to inhibit ER–endolysosomal interactions. However, Rab7 also interacts with FYCO1 (FYVE and coiled-coil domain-containing protein 1) and can mediate plus end-directed movement of the late endosomes/lysosomes [54]. Therefore, there may exist other mechanisms to fine-tune ER–endolysosomal interactions. Our study demonstrated that RILP binds to ORP1L to interfere with the formation of the ORP1L–VAP contact site, resulting in the clustering of the endolysosomes and accumulation of cholesterol in a Rab7-independent manner in fine-tuning ER–endolysosomal interactions (Figure 8).

RILP regulating cholesterol trafficking at the late endosomes and lysosomes has been investigated [25,55], but the mechanism remains elusive. The destination of the accumulated cholesterol in the late endosomes/lysosomes is not clear. In this study, we found that the overexpression of RILP inhibits ER–endolysosome interactions, consequently blocking the cholesterol export from the late endosomes/lysosomes. Further examination revealed that a high level of cholesterol in endolysosomes triggers autophagy, which is coincident with the previous investigation that high cholesterol induces autophagy [56,57], and consistent with the study that RILP participates in the autophagic pathway [58,59]. Our results indicated that a feed-back mechanism exists to protect the cell from stress-induced damage, namely the high level of cholesterol. The expression of RILP induces the accumulation of cholesterol, and the accumulated cholesterol will trigger autophagic degradation to clean excessive cholesterol (Figure 8). However, the patho-physiological consequences of RILP-induced cholesterol accumulation deserve further investigation.

## Figures and Tables

**Figure 1 cells-13-01313-f001:**
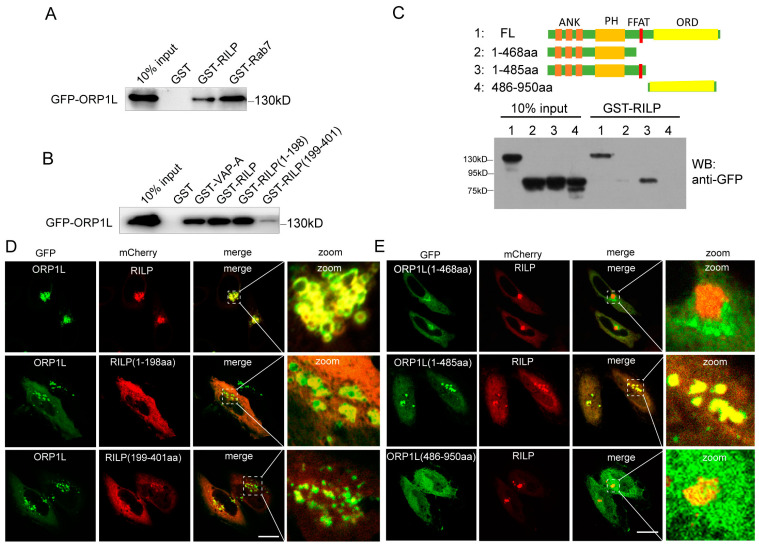
RILP interacts with ORP1L. (**A**) 293T cell lysates containing GFP-ORP1L were subjected to GST-pulldown assays using GST, GST-RILP, or GST-Rab7. The bound ORP1L to GST or GST fusion proteins was analyzed by Western blot using a GFP antibody. (**B**) 293T cell lysates containing GFP-ORP1L were subjected to GST-pulldown assays. The bound ORP1L to GST, GST-VAPa(1-225aa), GST-RILP, GST-RILP(1-198aa), or GST-RILP(199-401aa) was analyzed by Western blot using a GFP antibody. (**C**) 293T cell lysates containing GFP-ORP1L, GFP-ORP1L(1-468aa), GFP-ORP1L(1-485aa), or GFP-ORP1L(486-950aa) were subjected to GST-pulldown assays using GST-RILP. The bound ORP1L or the truncated mutants to GST-RILP were analyzed by Western blot using a GFP antibody. (**D**) Hela cells were co-transfected with GFP-ORP1L and mCherry-RILP, mCherry-RILP(1-198aa), or mCherry-RILP(199-401aa), and immunofluorescence microscopy revealed that GFP-ORP1L co-localized with mCherry-RILP and mCherry-RILP(1-198aa). (**E**) Hela cells were co-transfected with mCherry-RILP and GFP-ORP1L, GFP-ORP1L(1-468aa), GFP-ORP1L(1-485aa), or GFP-ORP1L(486-950aa), respectively. Immunofluorescence microscopy revealed that mCherry-RILP co-localized with ORP1L or ORP1L(1-485aa). Bar = 20 μm.

**Figure 2 cells-13-01313-f002:**
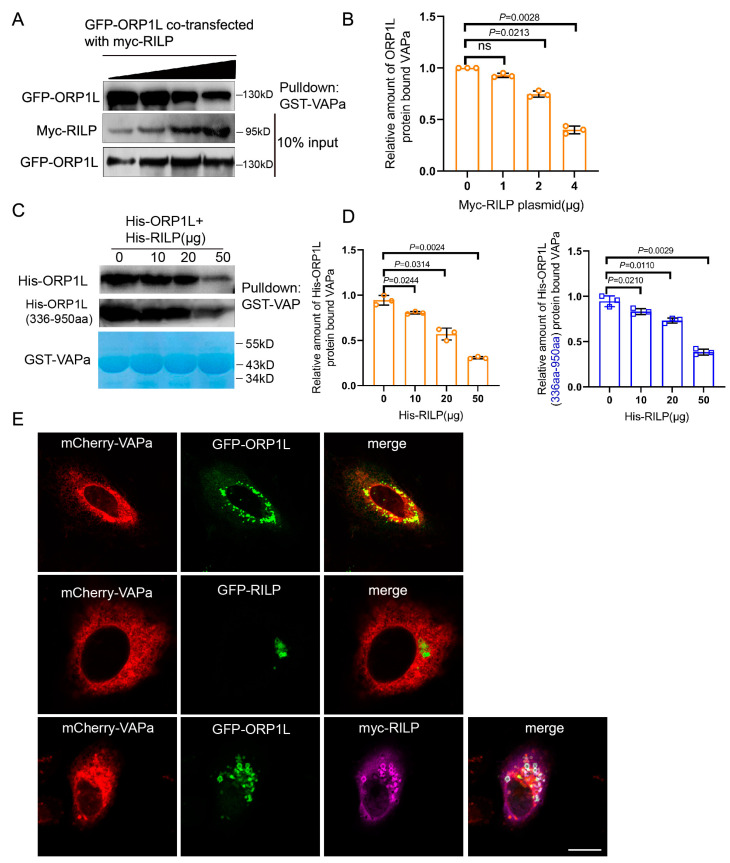
RILP competitively inhibits ORP1L interaction with VAPa. (**A**) GFP-ORP1L co-transfected with varied amounts of myc-RILP into Hela cells. The cell lysates were processed for GST-pulldown assays using GST-VAPa(1-225aa), and a Western blot showed that the amount of ORP1L binding to VAPa reduced with increasing amounts of RILP. (**B**) Quantitative analysis of the results of A from 3 independent experiments. ns, not significant. (**C**) The prokaryotic expressed GST-VAPa was used to bind His-ORP1L or His-ORP1L(336-950aa) by adding increasing amounts of His-RILP in vitro. Western blot demonstrated that RILP influences VAPa binding to either ORP1L or ORP1L(336-950aa). (**D**) Quantitative analysis of the results of C from 3 independent experiments. (**E**) Hela cells were co-transfected with GFP-ORP1L and mCherry-VAPa (upper panels), mCherry-RILP (middle panels), respectively, or GFP-ORP1L co-transfected with mCherry-VAPa (upper panels) and myc-RILP (lower panels). The myc-tag antibody was used to reveal myc-RILP. Cells were processed for immunofluorescence microscopy. Bar = 20 μm.

**Figure 3 cells-13-01313-f003:**
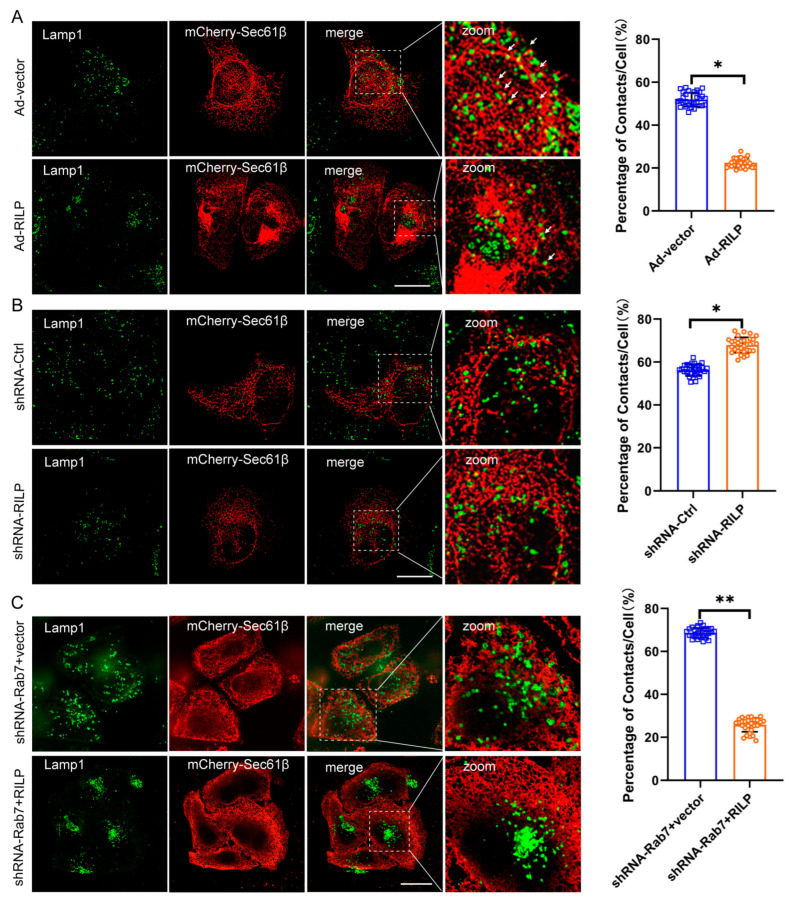
RILP interferes with ER–late endosome/lysosome contact. (**A**) Hela cells were infected with recombinant adenovirus expressing RILP (Ad-RILP) then transfected with mCherry-Sec61β. The late endosomes and lysosomes were labeled by Lamp1. Cells were processed for immunofluorescence microscopy by using a high intelligent and sensitive structured illumination microscope (His-SIM). A total of 30 cells were observed. ER–late endosome/lysosome contacts were analyzed by ImageJ_v1.8.0ImageJ. Group Ad-vector vs. Ad-RILP, * *p* < 0.0001, *t*-tests. (**B**) Hela cells were infected with lentivirus expressing pSicoR-shRNA-RILP. Cells were processed for immunofluorescence microscopy by His-SIM. ER–late endosome/lysosome contacts in 30 cells were analyzed by ImageJ. (**C**) Hela cells were transfected with lentivirus expressing pSicoR-shRNA-Rab7 and then infected with Ad-RILP. RILP mediates ER–late endosome/lysosome interactions in a Rab7-independent manner. ** *p* < 0.01, * *p* < 0.05, *t* tests. Bar = 20 μm.

**Figure 4 cells-13-01313-f004:**
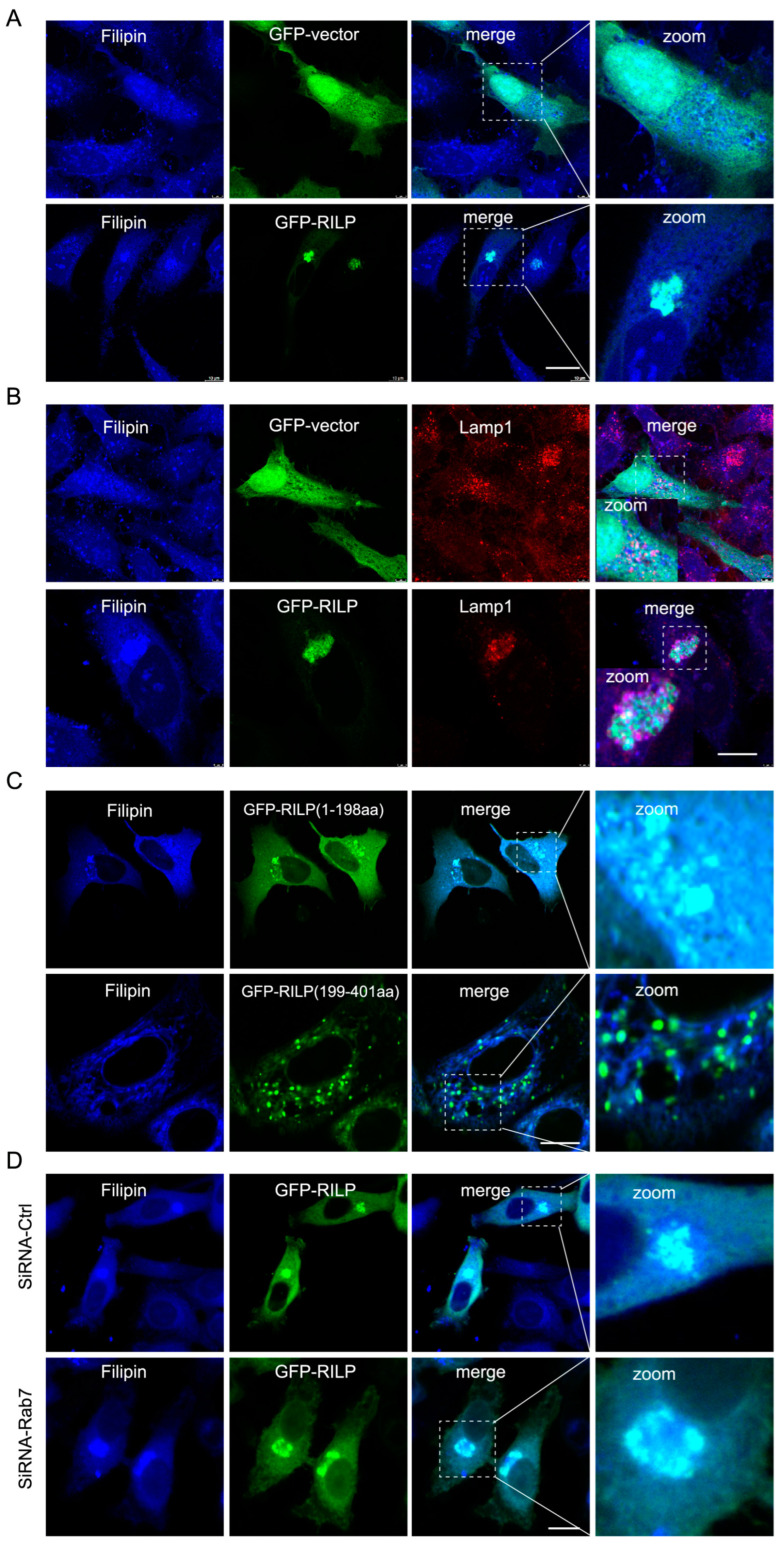
RILP induces cholesterol accumulation in the late endosomes/lysosomes. (**A**) Hela cells were transfected with GFP-RILP or GFP vectors, and cholesterol was labeled with Filipin. Cells were processed for immunofluorescence microscopy analysis. (**B**) Hela cells were transfected with GFP-RILP or GFP vectors, cholesterol was labeled with Filipin, and the late endosomes/lysosomes were labeled by Lamp1. (**C**) Hela cells were transfected with GFP-RILP(1-198aa) or GFP-RILP(199-401aa), and cholesterol was labeled with Filipin. (**D**) Hela cells were co-transfected with lentivirus expressing pSicoR-shRNA-Rab7 and GFP-RILP, labeling cholesterol with Filipin. Bar = 20 μm.

**Figure 5 cells-13-01313-f005:**
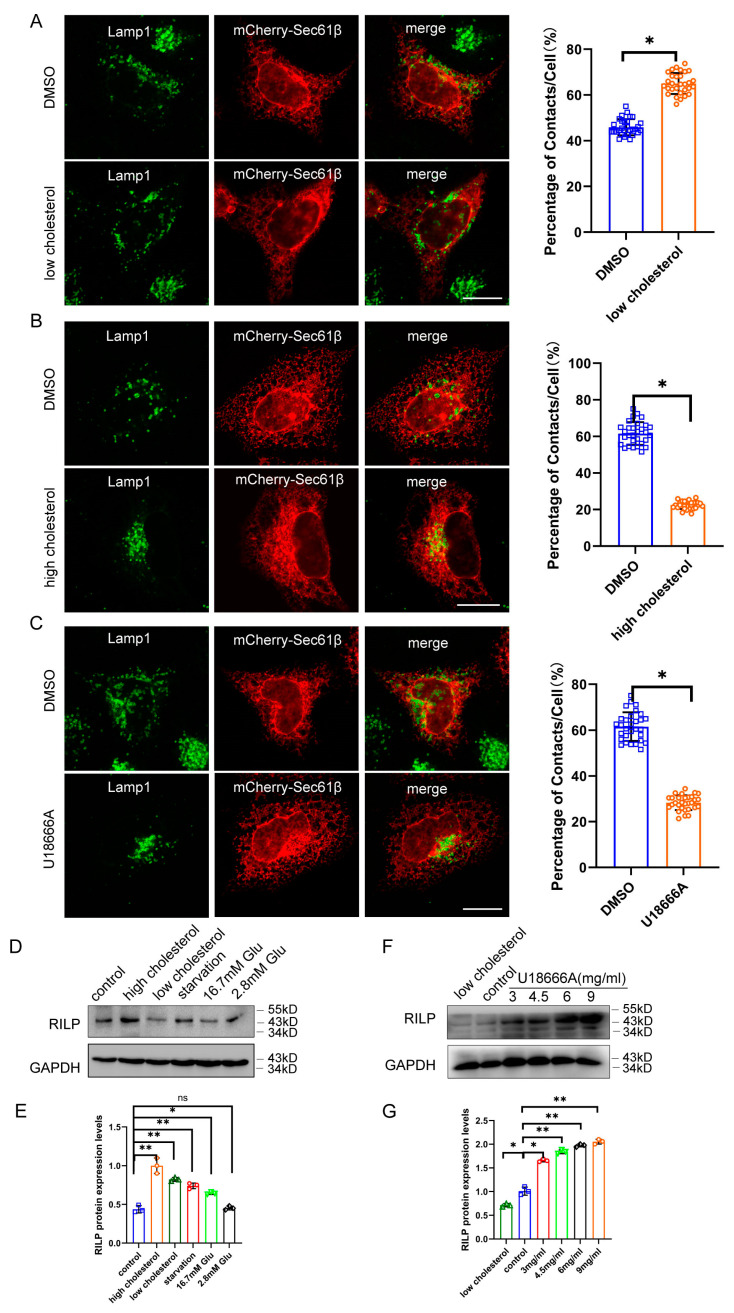
Accumulation of cholesterol in the endolysosome inhibits ER–endolysosomal contact. (**A**) Hela cells expressing mCherry-Sec61β were cultured in DMEM supplemented with cholesterol and immuno-labeled with Lamp1. Cells were processed for high-resolution immunofluorescence microscopy analysis. ER–Endolysosome contact in 30 cells were analyzed by ImageJ. (**B**) Hela cells expressing mCherry-Sec61β were cultured in DMEM supplemented with 5% lipoprotein-deficient serum, 230 μM mevalonate, and 50 μM lovastatin, then labeled with Lamp1. Cells were processed for high-resolution immunofluorescence microscopy analysis. ER–Endolysosome contact was analyzed by ImageJ. (**C**) Hela cells expressing mCherry-Sec61β were treated with 3 μg/mL U18666A for 15 h in DMEM, then labeled with Lamp1. Cells were processed for high-resolution immunofluorescence microscopy analysis. ER–Endolysosome contact was analyzed by ImageJ. Bar = 20 μm. (**D**) Hela cells were cultured under starvation, high- or low-cholesterol conditions, and 2.8 mM or 16.7 mM glucose conditions. Western blot was used to detect the protein levels of RILP. (**E**) Quantitative analysis of the results of (**D**) from 3 independent experiments. ns, not significant. (**F**) Hela cells were treated with varied amounts of U18666A. Western blot was used to detect the protein levels of RILP. (**G**) Quantitative analysis of the results of (**F**) from 3 independent experiments. ** *p* < 0.01, * *p* < 0.05, *t* tests.

**Figure 6 cells-13-01313-f006:**
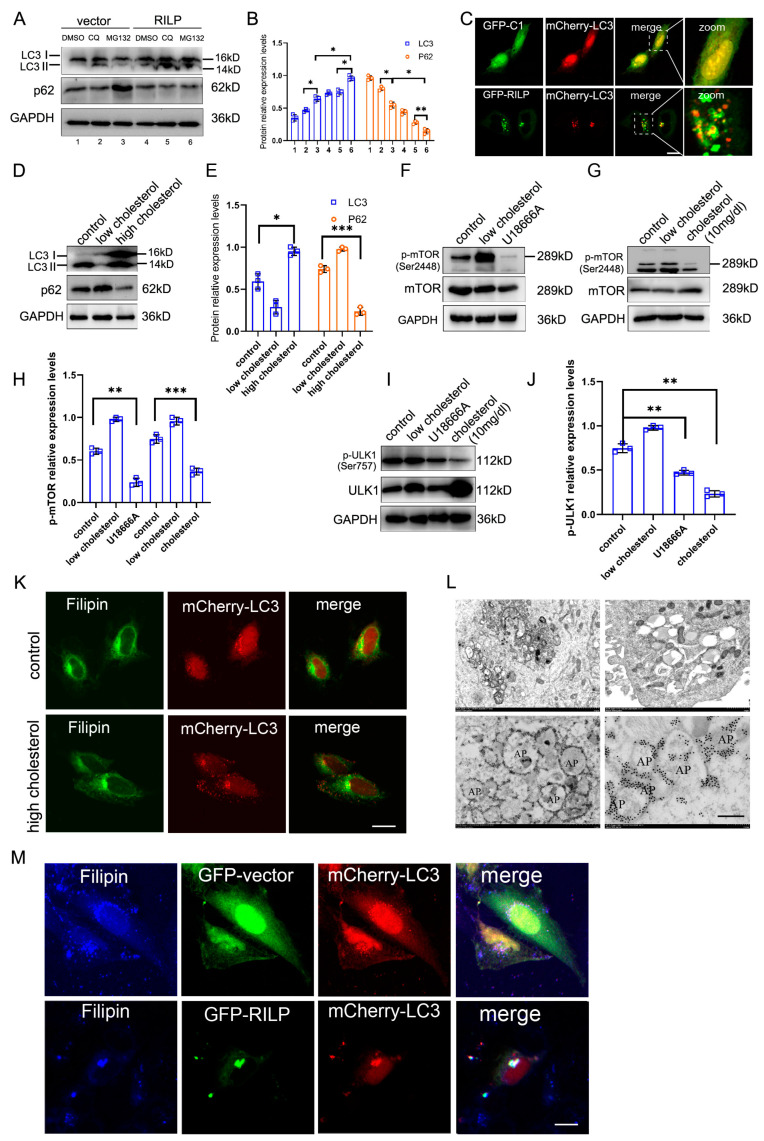
RILP-induced accumulation of cholesterol triggers autophagy. (**A**) Hela cells were infected with Ad-vector or Ad-RILP, then treated with chloroquine or MG132. The protein levels of LC3II were analyzed by Western blot. (**B**) Quantitative analysis of the results of A from 3 independent experiments. (**C**) Immunofluorescence microscopy revealed that GFP-RILP induces LC3-containing compartment clustering and LC3 co-localizes with RILP in Hela cells. (**D**) Western blot showed that high cholesterol significantly upregulates the expression of LC3II compared with low-cholesterol treatment. (**E**) Quantitative analysis of the results of D from 3 independent experiments**.** (**F**,**G**) Hela cells grew in DMEM with low or high concentrations of cholesterol or were treated with U18666A. Then, p-mTOR (Ser2448) was detected by Western blot. (**H**) Quantitative analysis of the results of (**F**,**G**) from 3 independent experiments. (**I**) Hela cells grew in DMEM with a high concentration of cholesterol or were treated with U18666A. Then, p-ULK1 (Ser757) was detected by Western blot. (**J**) Quantitative analysis of the results of I from 3 independent experiments. (**K**) Immunofluorescence microscopy revealed that high cholesterol increases LC3-containing particles (autophagosomes) in Hela cells. (**L**) Hela cells with lentivirus-mediated expression of RILP were analyzed by Transmit electronic microscopy, showing clustering of autophagosomes (upper panels) and the distribution of RILP (lower panels). (**M**) Hela cells were co-transfected with GFP-RILP and mCherry-LC3 and stained with Filipin. Immunofluorescence microscopy demonstrated that RILP induces cholesterol (marked by Filipin) accumulation and co-localization with autophagosomes. *** *p* < 0.001, ** *p* < 0.01, * *p* < 0.05, *t*-tests. Bar = 20 μm.

**Figure 7 cells-13-01313-f007:**
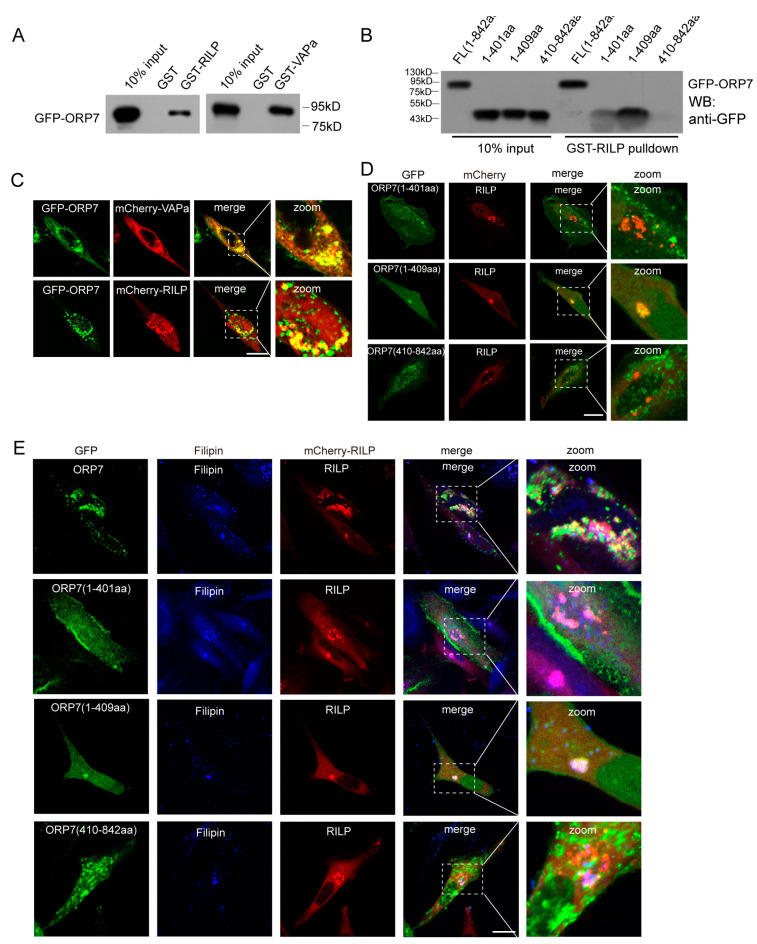
RILP interacts with the ORP family to regulate cholesterol trafficking. (**A**) 293T cell lysates containing GFP-ORP1L were subjected to GST-pulldown assays using GST, GST-RILP, or GST-VAPa(1-225a). The bound ORP1L to GST or the GST fusion protein was analyzed by Western blot using a GFP antibody. (**B**) 293T cell lysates containing GFP-ORP7, GFP-ORP7(1-401aa), GFP-ORP7(1-409aa), or GFP-ORP1L(410-842aa) were subjected to GST-pulldown assays using GST-RILP, the bound ORP7, or the truncated mutants to GST-RILP and analyzed by Western blot using a GFP antibody. (**C**) Hela cells were co-transfected ORP7 with RILP or VAPa. Immunofluorescence microscopy revealed that ORP7 co-localizes with RILP and VAPa. (**D**) RILP was co-transfected with ORP7(1-401aa), ORP7(1-409aa), or ORP7(410-842aa) in Hela cells, respectively. Immunofluorescence microscopy revealed that RILP co-localizes with ORP7(1-409aa), but not ORP7(1-401aa) or ORP7(410-842aa). (**E**) RILP was co-transfected with ORP7(1-401aa), ORP7(1-409aa), or ORP7(410-842aa) in Hela cells, respectively, then labeled with Filipin to detect cholesterol distribution. Immunofluorescence microscopy showed that ORP7 co-localizes with filipin-marked cholesterol, and overexpression of RILP induces ORP7 or ORP7(1-409aa) and cholesterol clustering. Bar = 20 μm.

**Figure 8 cells-13-01313-f008:**
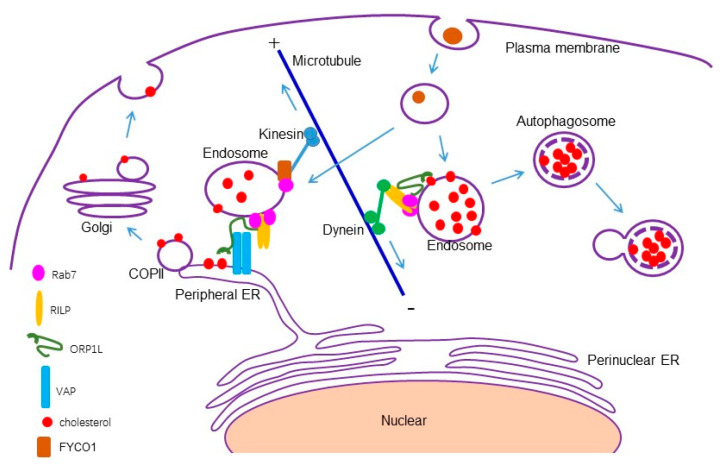
A model for RILP in regulating cholesterol trafficking. RILP interacts with ORP1L to competitively interfere with the formation of the ORP1L–VAP contact site, consequently inhibiting ER–endolysosome interactions and inhibiting cholesterol flow from the late endosomes/lysosomes to the endoplasmic reticulum, triggering autophagy through feedback.

## Data Availability

All relevant data are within the paper and its Appendix A.

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
