# Peer review of "RILP Induces Cholesterol Accumulation in Lysosomes by Inhibiting Endoplasmic Reticulum–Endolysosome Interactions"

_cells, 2024, doi:10.3390/cells13161313_

Round 1

Reviewer 1 Report

Comments and Suggestions for Authors

Endoplasmic reticulum (ER)-endolysosome interaction, mediated by VAP-ORP1L contact sites regulated by Rab7, facilitates cholesterol exchange between these organelles. This study found that RILP interacts with ORP1L to inhibit VAP-ORP1L contact site formation, causing endosome/lysosome clustering, reducing ER-lysosome contacts, and leading to cholesterol accumulation in lysosomes, which triggers autophagy. Although the mechanisms by which RILP precisely interferes with the ER-lysosome interaction are not well illustrated, this study provides significant insights into the regulation of ER-endolysosome interactions and cholesterol exchange. Nevertheless, additional details are necessary to enhance their findings, and a revision is required before further consideration.

Did the authors examine whether the knockdown of RILP upon high cholesterol treatment blocks the peri-nuclear clustering of late endosomes/lysosomes and autophagy? This experiment is crucial to determine the role of RILP in cholesterol metabolism and its potential as a therapeutic target in health and disease.

The author showed that “RILP(1-198aa) co-localized with ORP1L much more than RILP(199-401aa)”. However, it is difficult to distinguish this weak phenotype based on the low-quality images presented in the figure 1D. High-resolution images are needed to accurately represent this finding.

In Figures 3A and 3B, the LAMP1 images in the Ad-RILP and shRNA-control groups showed diverse distribution compared to the LAMP1 signal in Figure 4. It is unclear if this represents the true LAMP1 signal. Better quality images are needed for accurate representation.

In Figure 5, the percentages of contacts per cell in DMSO conditions showed large variation among A, B, and C. It is important to clarify whether these counts were performed using different methods. If not, the author should ensure consistency in these numbers.

In Figures 5E and 5G, the y-axis should be labeled as “…protein expression levels”. Additionally, statistical significance between the control and other groups should be performed and indicated.

Author Response

Comments and Suggestions for Authors

Endoplasmic reticulum (ER)-endolysosome interaction, mediated by VAP-ORP1L contact sites regulated by Rab7, facilitates cholesterol exchange between these organelles. This study found that RILP interacts with ORP1L to inhibit VAP-ORP1L contact site formation, causing endosome/lysosome clustering, reducing ER-lysosome contacts, and leading to cholesterol accumulation in lysosomes, which triggers autophagy. Although the mechanisms by which RILP precisely interferes with the ER-lysosome interaction are not well illustrated, this study provides significant insights into the regulation of ER-endolysosome interactions and cholesterol exchange. Nevertheless, additional details are necessary to enhance their findings, and a revision is required before further consideration.

R: Thanks for your encouraging comments and suggestions. 

Did the authors examine whether the knockdown of RILP upon high cholesterol treatment blocks the peri-nuclear clustering of late endosomes/lysosomes and autophagy? This experiment is crucial to determine the role of RILP in cholesterol metabolism and its potential as a therapeutic target in health and disease.

R: Thank you for your good suggestion and insightful comment. In this stage, we didn’t examined whether RILP knockdown blocks the peri-nuclear clustering of late endosomes/lysosomes and autophagy upon high cholesterol treatment. We agree that this experiment is crucial to determine the role of RILP in cholesterol metabolism and its potential as a therapeutic target in health and disease. Based on our present data, RILP-Knockdown did result in more peripheral distribution of the late endosomes/lysosomes (Figure 3) and more peripheral distribution of cholesterol (Figure S2B), suggesting depletion of RILP will block the perinuclear clustering of cholesterol-loaded late endosomes/lysosomes. The influence of RILP-knockdown on cholesterol-triggered autophagy deserves further investigations. 

The author showed that “RILP(1-198aa) co-localized with ORP1L much more than RILP (199-401aa)”. However, it is difficult to distinguish this weak phenotype based on the low-quality images presented in the figure 1D. High-resolution images are needed to accurately represent this finding.

R: Thank you for your comments. Indeed, both RILP(1-198aa) and RILP(199-401aa) are more cytosolic with punctual structures disperses throughout the cytoplasm. In our observations, ORP1L co-localizes with the most puncta of RILP(1-198aa), ORP1L also will co-localize with some puncta of RILP(199-401aa). Since both the puncta structures are endosomal structures, thus immunofluorescence microscopy may not distinguish them. In order to see more clearly, we modified the Figure 1D and 1E by adding an amplification to show the co-localization.  

In Figures 3A and 3B, the LAMP1 images in the Ad-RILP and shRNA-control groups showed diverse distribution compared to the LAMP1 signal in Figure 4. It is unclear if this represents the true LAMP1 signal. Better quality images are needed for accurate representation.

R: Thanks for your comments. The data of Figures 3A and 3B were constructed with multilayer pictures obtained through high intelligent and sensitive structured illumination microscope (His-SIM), so you can see more the peripheral late endosomes/lysosomes. The data of Figure 4B was obtained with normal confocal microscope, focusing on the representative distribution of lysosomes on mono-layer. It is appreciated for your understanding this divergence.  

In Figure 5, the percentages of contacts per cell in DMSO conditions showed large variation among A, B, and C. It is important to clarify whether these counts were performed using different methods. If not, the author should ensure consistency in these numbers.

R: Thanks for your comments and suggestion.Because the experiments for Figure5A, 5B and 5C were carried out, respectively, using different batches of cells, so there exists variation in statistical analysis among 5A, 5B and 5C. In order to show more clearly, we modified the statistical analysis results by applying scatter plots (30 cells were observed) in statistical analysis, and the statistical comparison results have been described in the figure legend, adding the appropriate precise p-values.

In Figures 5E and 5G, the y-axis should be labeled as “…protein expression levels”. Additionally, statistical significance between the control and other groups should be performed and indicated.

R: Thanks for your suggestions.We have modified the y-axis labels in Figures 5E and 5G. And as suggested, Statistical significance between the control and other groups was assessed.

Reviewer 2 Report

Comments and Suggestions for Authors

In the submitted Manuscript by Han et al. the authors analyze the role of the Rab7-interacting lysosomal protein (RILP) in the interaction between the endoplasmatic reticulum (ER) and endolysosomes. They show that RILP interacts with ORP1L through binding to its FFAT motif and thereby preventing the formation of ORP1L-VAP contact sites between ER and endolysosomes, which inhibits the interaction between ER and endolysosomes. This results in accumulation of cholesterol in late endosomes, which induces autophagy. The manuscript has a clear and plausible structure, is written in an understandable manner and includes most relevant references concerning the topic of the manuscript. The overall quality of the manuscript (including the figures) is very good and the conclusions, which are made by the authors, are supported by the provided data. The size of the manuscript, which contains 8 main figures, is also quite extensive, therefore I think no further experiments need to be done. Due to this, I have only a few minor points that should be edited before the manuscript can be published in the Cells journal.    

Minor points:

Abstract:  All abbreviations (e.g. Rab7, RILP, VAP, ORP1L) should be explained by spelling them out, when they are used for the first time and then the abbreviations can be used for subsequent repetitions. In addition, even if the same abbreviation is used in the abstract several times, it should be spelt out again, when it is used for the first time in the body part of the manuscript.

Page 3, line 110: There is no need to cite this publication (26) at this place. These are standard culture conditions with standard medium and temperature, therefore there is no need to include a reference here.

The reference #26 is cited again on page 4, line 168, but here the citation is adequate because it refers to a technique, which was used in a previous work. However, on page 3, line 110 the reference should be removed, since it is a publication from the same research group and therefore a wrong impression of excessive self-citation can be created.

Microscopy pictures: The authors should include an additional labeling into all microscopy figures for the overlay of different channels. For example, in Figure 1D there is a labeling for the green channel: “GFP” and for the red channel: “mCherry”. The overlay images can be labeled as “overlay” or “merge”.

The inserts (detailed magnification of some areas) of the images should also be labeled as “insert” or “zoom” and it should be indicated with a dashed line box, which part of the image is shown in detail as an insert. This applies to figures 3A-C, 4A-D, 6C and 7C-E

In addition, if the authors use microscopy pictures where they show co-localization or no co-localization of different proteins (e.g. Figure 1E), they should include an additional graph showing    line-scan profiles of the fluorescence intensity of the different channels, which would make it easier to see, whether there is co-localization or not. An example for such a line scan profile can be found here: https://www.researchgate.net/figure/top-CLSM-images-and-bottom-line-scan-profiles-of-fluorescence-intensity-for-Hela_fig2_221897258

Figure 2B and 2E: The error bars in the bar charts indicate that the experiment was performed more than once, therefore the statistics (P-value) can be shown by the authors (e.g. t-test).

Figure 3A: Lamp1 should be mentioned somewhere in the text on page 9 (that it is located on endosomes/lysosomes) so that it is clear to the reader that Lamp1 staining was done in order to identify lysosomes.

Author Response

Responses to Reviewer 2 

In the submitted Manuscript by Han et al. the authors analyze the role of the Rab7-interacting lysosomal protein (RILP) in the interaction between the endoplasmatic reticulum (ER) and endolysosomes. They show that RILP interacts with ORP1L through binding to its FFAT motif and thereby preventing the formation of ORP1L-VAP contact sites between ER and endolysosomes, which inhibits the interaction between ER and endolysosomes. This results in accumulation of cholesterol in late endosomes, which induces autophagy. The manuscript has a clear and plausible structure, is written in an understandable manner and includes most relevant references concerning the topic of the manuscript. The overall quality of the manuscript (including the figures) is very good and the conclusions, which are made by the authors, are supported by the provided data. The size of the manuscript, which contains 8 main figures, is also quite extensive, therefore I think no further experiments need to be done. Due to this, I have only a few minor points that should be edited before the manuscript can be published in the Cells journal.    

R: Many thanks for your encouraging comments and suggestions. 

Minor points: 

Abstract: All abbreviations (e.g. Rab7, RILP, VAP, ORP1L) should be explained by spelling them out, when they are used for the first time and then the abbreviations can be used for subsequent repetitions. In addition, even if the same abbreviation is used in the abstract several times, it should be spelt out again, when it is used for the first time in the body part of the manuscript.

R: Thank you for your suggestion.We have modified the descriptions for the major abbreviations (such as Rab7, RILP, VAP, ORP1L, ER, EGFR, MSP, TMCC1, PTP1B, VPS13, RNF26, PDZ8 and IP3R) in the abstract  and the main text.

 Page 3, line 110: There is no need to cite this publication (26) at this place. These are standard culture conditions with standard medium and temperature, therefore there is no need to include a reference here.R: Thank you for your suggestion. The references has been removed. The reference #26 is cited again on page 4, line 168, but here the citation is adequate because it refers to a technique, which was used in a previous work. However, on page 3, line 110 the reference should be removed, since it is a publication from the same research group and therefore a wrong impression of excessive self-citation can be created.

R: Thanks for your suggestion. The reference cited here has been removed. 

Microscopy pictures: The authors should include an additional labeling into all microscopy figures for the overlay of different channels. For example, in Figure 1D there is a labeling for the green channel: “GFP” and for the red channel: “mCherry”. The overlay images can be labeled as “overlay” or “merge”.

R: Thanks for your suggestion. Overlay images in all microscope images have been labeled as “merge”. 

The inserts (detailed magnification of some areas) of the images should also be labeled as “insert” or “zoom” and it should be indicated with a dashed line box, which part of the image is shown in detail as an insert. This applies to figures 3A-C, 4A-D, 6C and 7C-E

R: Thank you for your suggestion. The inserts (detailed magnification of some areas) in figures 3A-C, 4A-D, 6C, and 7C-E have been marked and indicated by dashed as “zoom”. 

In addition, if the authors use microscopy pictures where they show co-localization or no co-localization of different proteins (e.g. Figure 1E), they should include an additional graph showing line-scan profiles of the fluorescence intensity of the different channels, which would make it easier to see, whether there is co-localization or not. An example for such a line scan profile can be found here: https://www.researchgate.net/figure/top-CLSM-images-and-bottom-line-scan-profiles-of-fluorescence-intensity-for-Hela_fig2_221897258.

R: Thanks for your suggestion. We agree that line-scan profiles of the fluorescence intensity will provide additional information for fluorescent signals, however, the overlay images must be enough to show the co-localization in our experiments. In order to see more clearly, we modified the correspondent Figures by adding the amplification area to show the co-localization. 

Figure 2B and 2E: The error bars in the bar charts indicate that the experiment was performed more than once, therefore the statistics (P-value) can be shown by the authors (e.g. t-test).

R: Thanks for your suggestion.We modified the figures by applying scatter plots to the charts, and the appropriate precise p-values were added in the figure legend.

 Figure 3A: Lamp1 should be mentioned somewhere in the text on page 9 (that it is located on endosomes/lysosomes) so that it is clear to the reader that Lamp1 staining was done in order to identify lysosomes.

R: Thanks for your suggestion. We have revised the description, indicating the endosomes/lysosomes were marked by Lamp1 in page 8 and 9.

Reviewer 3 Report

Comments and Suggestions for Authors The manuscript of Han et al. tells a very interesting story about how RILP (the Rab7 adapter protein that links Rab7 to dynein/dynactin) can bind to ORP1L (the oxysterol binding protein like protein 1L) and prevents its binding to VAPa (the ER resident protein).  These interactions were well-established by co-immunoprecipitation with full-length and truncation mutants.  The data was supported by fluorescence micrographs of partial colocalization.  The consequence of RILP binding is the movement of late endosomes/lysosomes to a perinuclear location and the prevention of unloading of cholesterol from the late endosomes/lysosomes. The authors postulate that the accumulation of cholesterol in the late endosomes/lysosomes triggers autophagy and that this condition may involve the recruitment of other members of the ORP family including ORP7.  This is an interesting and important work, especially the demonstration of the functional importance of RILP competing with VAPa binding to ORP1L.  There are many grammar mistakes that detract from the message.  I find the fluorescence microscopy to be sloppy and non-quantitative, and thus poorly supporting the excellent co-immunoprecipitation data.  Furthermore, I think the authors try to do too much.  They have some excellent data which demonstrates that RILP competes with VAPa for binding to ORP1L and this prevents proper unloading of cholesterol from the late endosomes/lysosomes.  I find the induction of autophagy (Figure 6) and the implication of ORP7 (Figure 7) to be less well explained and confuses a very good story.  I would suggest withholding the data from Figure 6 and 7 and focussing on improving the quality and "quantifiability" of the data.  My comments are below.   
  1. The interesting role of Rab7 in late endosome/lysosome trafficking is beautiful, but complicated.  Sometimes, it involves a "tug-of-war" between FYCO/kinesin and RILP/dynein/dynactin to determine the intracellular localization of the late endosomes/lysosomes.  While I find a lot of the results about the role of RILP to be compelling, the authors missed the other half of the equation about what is happening to FYCO/kinesin.  While I am sure the authors could fill another entire manuscript with experiments on FYCO/kinesin, the complete disregard to FYCO/kinesin was also not helpful.  Does FYCO decrease in the presence of RILP? Does FYCO promote binding to ORP1L?  This certainly would have enhanced the overall understanding.  In this way, Figure 8 would make sense. This model is incoherent in its current form.  
  2. Lysosomal localization is very important.  The work on mTORC has shown this very well and the authors contribution is valuable to this discussion.  However, Figure 8 does not do a good job of conveying this.  The left hand side is showing COPII dependent distribution of free cholesterol from the ER?  This model should be showing peripheral and perinuclear ER, microtubule motors, RILP/FYCO/Rab7 competing to deliver the late endosomes/lysosomes either peripherally or perinuclearly.  Upon overexpression of RILP, (-)-ended motors win the competition and cluster the late endosomes/lysosomes perinuclearly, where ORP1L is recruited, preventing binding to VAPa through ORP1L's FFAT domain.  As a result, incomplete cholesterol mobilization from the late endosomes/lysosomes occurs.  The model does not make it clear what happens to the late endosomes/lysosomes and the cholesterol within, showing some unusual transformation into an autophagosome.  What happens to the cholesterol then?  So, a lot of details are missing here.  I am sure the authors can do a better job.
  3. Overall, I find the imaging to be less convincing that the other experiments.  The authors have selected one image to display their premise.  How many cells were viewed?  Some of the fluorophores are spread throughout the cell and so the chance of "co-localization" with another fluorophore is very high, but likely not physiological (at least their colocalization is not indicative of a functional interaction).  Furthermore, colocalization is a relative term, that can be because of coincidence, without inferring any functional significance.  The fluorophores are often so strong in intensity and widespread that "colocalization" is inevitable (eg. Fig 3ABC).  I often don't know what colour I should be looking for (yellow, white?) as it is not stated in the Figure legend what I should be looking for.  Overall, this is sloppy fluorescence microscopy.  
  4. Specifically, for example in Figure 1D, I would want to see ORP1L alone without expression of RILP. In Figure 1DE, I would want to see blown up images of the "co-localization" events.  Figure 4: filipin staining has not been done properly or one would expect that the plasma membranes to be highlighted as this is where the vast majority of the cellular free cholesterol is.  
  5. What is the location of VAPa?  It is on the ER but does it have peripheral or perinuclear localization? The authors have used some very nice data to suggest that RILP promotes perinuclear localization of late endosomes/lysosomes and thereby prevents the interaction between ORP1L and VAPa.  From the images in Figure 2E, VAPa has both peripheral and perinuclear localization.  How does this fit with your hypothesis?
  6. What is the location of ORP1L normally?  It has an FFAT domain, so we would expect it to colocalize with the VAPa in the ER.  Correct?  Figure 1D is a bit confusing then, as I would expect ORP1L to be peripherally located in the absence of RILP (or a mutant lacking the FFAT domain)?
  7. The authors claim that cholesterol accumulates in the late endosomes/lysosomes.  The authors have loosely used the term cholesterol and it is not clear if they mean both free cholesterol and cholesteryl ester.  Have the authors, for example, created a lysosomal storage condition/disease by overexpressing RILP? The filipin staining is unconvincing (Figure 4) and should be staining the plasma membrane as well as some internal compartments. (See for examples, https://doi.org/10.1172/JCI8087. or https://www.sciencedirect.com/science/article/pii/S0091679X14000296). It might be helpful to stain the cells with BODIPY/Nile Red and colocalize with lysosomal markers or Lysotracker type reagents to prove the colocalization of cholesteryl ester in the late endosomes/lysosomes. 
  8. Figure 6: In western blots with multiple bands (figures 6A, D, G), please indicate where the proposed band is located. For example, Figure 6A, where is LC3I and LC3II? I am also confused about the interpretation of these results.  Does more LC3II and p63 indicate more autophagy or less flux?  Figures 6KLM are not at all convincing of how the cholesterol enriched late endosomes/lysosomes and the autophagosomes are related.  Are the authors suggesting that the cholesterol-enriched late endosomes are engulfed within autophagosomes?  
  9. I am concerned that introduction of ORP7 confuses the issues and dilutes the message of the manuscript.  While the observation is valid and could be very important (very little is known about ORP7), the authors have thrown in this observation without fully elucidating what this means.  I would prefer the authors to remove this figure and save it for another manuscript.
Comments on the Quality of English Language

There are lots of grammar mistakes.  A simple read through by an English speaker should fix the problems.

Author Response

Responses to Reviewer 3

The manuscript of Han et al. tells a very interesting story about how RILP (the Rab7 adapter protein that links Rab7 to dynein/dynactin) can bind to ORP1L (the oxysterol binding protein like protein 1L) and prevents its binding to VAPa (the ER resident protein).  These interactions were well-established by co-immunoprecipitation with full-length and truncation mutants.  The data was supported by fluorescence micrographs of partial colocalization.  The consequence of RILP binding is the movement of late endosomes/lysosomes to a perinuclear location and the prevention of unloading of cholesterol from the late endosomes/lysosomes. The authors postulate that the accumulation of cholesterol in the late endosomes/lysosomes triggers autophagy and that this condition may involve the recruitment of other members of the ORP family including ORP7.  This is an interesting and important work, especially the demonstration of the functional importance of RILP competing with VAPa binding to ORP1L.  There are many grammar mistakes that detract from the message.  I find the fluorescence microscopy to be sloppy and non-quantitative, and thus poorly supporting the excellent co-immunoprecipitation data.  Furthermore, I think the authors try to do too much.  They have some excellent data which demonstrates that RILP competes with VAPa for binding to ORP1L and this prevents proper unloading of cholesterol from the late endosomes/lysosomes.  I find the induction of autophagy (Figure 6) and the implication of ORP7 (Figure 7) to be less well explained and confuses a very good story.  I would suggest withholding the data from Figure 6 and 7 and focussing on improving the quality and "quantifiability" of the data.  My comments are below.  

R: Many thanks for your encouraging comments and suggestions. 

The interesting role of Rab7 in late endosome/lysosome trafficking is beautiful, but complicated.  Sometimes, it involves a "tug-of-war" between FYCO/kinesin and RILP/dynein/dynactin to determine the intracellular localization of the late endosomes/lysosomes.  While I find a lot of the results about the role of RILP to be compelling, the authors missed the other half of the equation about what is happening to FYCO/kinesin.  While I am sure the authors could fill another entire manuscript with experiments on FYCO/kinesin, the complete disregard to FYCO/kinesin was also not helpful.  Does FYCO decrease in the presence of RILP? Does FYCO promote binding to ORP1L?  This certainly would have enhanced the overall understanding.  In this way, Figure 8 would make sense. This model is incoherent in its current form.

R: Thanks for your insightful comments.The endosome/lysosome positioning is regulated by multiple machineries, Rab7 must be the master regulator for this event. As you mentioned, Rab7 also interacts with FYCO1/Kinesin, directing microtubule plus-end movement of the endosome/lysosome, which is opposite to Rab7/RILP/Dynein regulated movement, and we have discussed this finding in the discussion section. Interestingly, FYCO1 also regulates ER-endosome contact by interaction with protruding (Raiborg et al.2015,Nature). In this stage, we didn’t examine whether FYCO1 balances the RILP’s effect, especially under cholesterol manipulation. We think this would be another story. However, we have modified the model in Figure 8 showing a potential role of FYCO1/Kinesin.

 Lysosomal localization is very important.  The work on mTORC has shown this very well and the authors contribution is valuable to this discussion.  However, Figure 8 does not do a good job of conveying this.  The left hand side is showing COPII dependent distribution of free cholesterol from the ER?  This model should be showing peripheral and perinuclear ER, microtubule motors, RILP/FYCO/Rab7 competing to deliver the late endosomes/lysosomes either peripherally or perinuclearly.  Upon overexpression of RILP, (-)-ended motors win the competition and cluster the late endosomes/lysosomes perinuclearly, where ORP1L is recruited, preventing binding to VAPa through ORP1L's FFAT domain.  As a result, incomplete cholesterol mobilization from the late endosomes/lysosomes occurs.  The model does not make it clear what happens to the late endosomes/lysosomes and the cholesterol within, showing some unusual transformation into an autophagosome.  What happens to the cholesterol then?  So, a lot of details are missing here.  I am sure the authors can do a better job.

R: Thanks for your insightful comments and suggestionsWe have tried our best to modify Figure 8, it is hoped that you will find the revised version is better than the old one.

Overall, I find the imaging to be less convincing that the other experiments.  The authors have selected one image to display their premise.  How many cells were viewed?  Some of the fluorophores are spread throughout the cell and so the chance of "co-localization" with another fluorophore is very high, but likely not physiological (at least their colocalization is not indicative of a functional interaction).  Furthermore, colocalization is a relative term that can be because of coincidence, without inferring any functional significance.  The fluorophores are often so strong in intensity and widespread that "colocalization" is inevitable (eg. Fig 3ABC).  I often don't know what colour I should be looking for (yellow, white?) as it is not stated in the Figure legend what I should be looking for.  Overall, this is sloppy fluorescence microscopy.

R: Thank you for your comments.Yes, we agree that sometimes it is inevitable to see coincident co-localization, especially for cytosolic proteins which spread throughout the cytoplasm. However, membrane proteins, such as VAP and Lamp1, may specifically locate at the membrane structures, the colocalization between membrane proteins should be functionally significance. For proteins (eg. ORP1L and RILP) which associated with membrane through binding their partners, we can define the membrane associated pool and investigate their colocalization.

In Figure 3ABC, both VAP-A (ER marker) and Lamp1(LE/Lysosome marker) are membrane proteins, the yellow fluorophore is observed when the lsysosome contacts ER. Some representative contacts were indicated by white arrow in the figures. We have revised the analytic chart by applying scatter plots, and cell number observed were indicated the text and Figure legend.

Specifically, for example in Figure 1D, I would want to see ORP1L alone without expression of RILP. In Figure 1DE, I would want to see blown up images of the "co-localization" events.  Figure 4: filipin staining has not been done properly or one would expect that the plasma membranes to be highlighted as this is where the vast majority of the cellular free cholesterol is. 

R: Thank you for your suggestion and comments. Normally, ORP1L disperses in the cytoplasm around the nuclear, with major pool of ORP1L associates with the late endosome/lysosome marked by Lamp1. We added this data in supplemental Figure s1A, and indicated in the main text. In order to see more clearly, we modified the correspondent Figures by adding the amplification area to show the co-localization. The filipin staining in Figure 4 was performed according to literature reports (Chu et al., 2015, Cell).

.  What is the location of VAPa?  It is on the ER but does it have peripheral or perinuclear localization? The authors have used some very nice data to suggest that RILP promotes perinuclear localization of late endosomes/lysosomes and thereby prevents the interaction between ORP1L and VAPa.  From the images in Figure 2E, VAPa has both peripheral and perinuclear localization.  How does this fit with your hypothesis?

R: Thank you for your comments. VAPa is an ER membrane protein both in the peripheral and perinuclear ER. As discussed in text, peripheral tubular ER prefer to contact with other organelle, while perinuclear ER sheet not suitable for organelle-organelle interaction. The results in Figure 2E are consistent with our hypothesis, as RILP promotes perinuclear localization of the late endosomes/lysosomes and thereby prevents the interaction between ORP1L and VAPa. 

What is the location of ORP1L normally?  It has an FFAT domain, so we would expect it to colocalize with the VAPa in the ER.  Correct?  Figure 1D is a bit confusing then, as I would expect ORP1L to be peripherally located in the absence of RILP (or a mutant lacking the FFAT domain)?

R: Thank you for your comments. Normally, ORP1L associates the late endosomes/lysosome (Figure s1A added). The membrane association of ORP1L is regulated by multiple factors, N-terminal can bind ro Rab7, also the PH domain can bind to phosphatidyllipid on the membrane, therefore disruption of FFAT can’t completely abolish its endosomal membrane association.  

The authors claim that cholesterol accumulates in the late endosomes/lysosomes.  The authors have loosely used the term cholesterol and it is not clear if they mean both free cholesterol and cholesteryl ester.  Have the authors, for example, created a lysosomal storage condition/disease by overexpressing RILP? The filipin staining is unconvincing (Figure 4) and should be staining the plasma membrane as well as some internal compartments. (See for examples, https://doi.org/10.1172/JCI8087. or https://www.sciencedirect.com/science/article/pii/S0091679X14000296). It might be helpful to stain the cells with BODIPY/Nile Red and colocalize with lysosomal markers or Lysotracker type reagents to prove the colocalization of cholesteryl ester in the late endosomes/lysosomes.

R: Thank you for your suggestion and comments.The term cholesterol in this manuscript is defines as cholesterol, since it can be redistributed to other membrane structures through membrane trafficking.Unfortunately, we haven’t tried to creat a LSD model so far.The Filipin staining experiments were carried out as described (Chu et al., 2015, Cell). It appreciated very much for your understanding for this limitation. 

Figure 6: In western blots with multiple bands (figures 6A, D, G), please indicate where the proposed band is located. For example, Figure 6A, where is LC3I and LC3II? I am also confused about the interpretation of these results.  Does more LC3II and p63 indicate more autophagy or less flux?  Figures 6KLM are not at all convincing of how the cholesterol enriched late endosomes/lysosomes and the autophagosomes are related.  Are the authors suggesting that the cholesterol-enriched late endosomes are engulfed within autophagosomes? 

R: Thank you for your suggestion.The proposed bands have been indicated in the figure 6A,D,G. Elevation of LC3-II /LC3-I indicates the increase of autophagy. The expression level of p62 protein is negatively correlated with autophagic activity. We have modified the description in the main text.

Figure 6K showed a significant increase in the number of autophagosomes LC3 under high cholesterol conditions, indicating that high cholesterol promotes autophagy. Figures 6L and M demonstrated that RILP can promote an increase in the number of autophagosomes, and RILP can induce cholesterol accumulation in lysosomes and autophagosomes. Together with the Figure 6A-J, these results suggested that RILP mediated cholesterol-triggered autophagy. However, autophagosome formation and maturation is complicated, the late endosomes rich in cholesterol may be dysfunction and engulfed by autophagosomes.

I am concerned that introduction of ORP7 confuses the issues and dilutes the message of the manuscript.  While the observation is valid and could be very important (very little is known about ORP7), the authors have thrown in this observation without fully elucidating what this means.  I would prefer the authors to remove this figure and save it for another manuscript.

R: Thank you for your suggestion. We try to propose a common mechanism for RILP regulating cholesterol trafficking through interaction with ORP family proteins, thus this data is necessary for the manuscript. 

Comments on the Quality of English LanguageThere are lots of grammar mistakes.  A simple read through by an English speaker should fix the problems.

R: Thank you for your suggestion.We have tried our best to improve the English writing of the manuscript.

Round 2

Reviewer 3 Report

Comments and Suggestions for Authors The manuscript of Han et al. is a resubmission of a manuscript of the role of RILP on late endosomal/lysosomal positioning and the competition of RILP with ORP1L for VAP-A.  The manuscript is important and substantial.  The authors have addressed most of my concerns.  My comments are below.   Figure 2E remains a problem.  I believe the intention is to show that expression of RILP results in the reorganization of ORP1L.  What would be really helpful is to show that ORP1L moves from an ER compartment (first set of panels; highlighted by VAP-A colocalization, as shown) to a late endosomal/lysosomal compartment (here you need a Rab7/lysotracker marker).  It really looks like ORP1L is moving but there is only one way to prove this: show colocalization with a late endosomal/lysosomal marker!  This is where the fluorescence microscopy could be valuable.  It is also really helpful here to show statistics (100 cells counted, so many interactions without RILP, so many interactions with RILP), and calculate some kind of statistical significance.  Then, this would be really solid.  Otherwise it is speculation.   Line 300-305: I would be very careful about speculating that knocking down Rab7 has no effect on RILP localization.  I see no data showing the degree of Rab7 knockdown (western blot), as even a small amount of Rab7 may be sufficient.  On the other hand, there may be "hidden effects" of Rab7 knockdown that we do not see, that greatly affect the cells.  It is a leap of faith to suggest that other Rabs are involved based on this thin amount of data.     Figure 4B: there is no Zoom here.  This would be helpful.  I renew my objection to the filipin staining.  I gave the authors examples of what filipin staining looks like.  Also, there is no demonstration of how often this interaction occurs, or whether the authors have chosen the one picture that looks good.  There is no details of how many cells were counted here or of any statistics.   Furthermore, filipin stains free cholesterol.  That means, the authors are showing a perinuclear clustering of late endosomes/lysosomes with an accumulation of free cholesterol.  This is a lysosomal storage condition.     Figure 6: Although I still don't understand how late endosomes/lysosomes undergo autophagy, and then what happens to the free cholesterol. I accept that it appears to be true under the conditions and accept that the authors prefer to keep this figure.  Figure 6M is so small to be almost meaningless.   Figure 7: Although I still have some misgivings about the rationale for the choice to use ORP7, I accept the authors decision to keep this figure.   I would ask the authors to deliberately distinguish between total cholesterol, free cholesterol, and cholesteryl ester, the many times that it is mentioned in the text.  I still think the authors use the term cholesterol too loosely, as if they are all the same.     The supplemental files were in a .rar format and I could not access them.  Comments on the Quality of English Language

Just some minor errors.  For example: last line of abstract

"Our results suggest that RILP interferes ER-endolysosome interaction " should be

"Our results suggest that RILP interferes with ER-endolysosome interaction..."

Author Response

Comments and Suggestions for Authors

The manuscript of Han et al. is a resubmission of a manuscript of the role of RILP on late endosomal/lysosomal positioning and the competition of RILP with ORP1L for VAP-A.  The manuscript is important and substantial.  The authors have addressed most of my concerns.  

My comments are below.  

Figure 2E remains a problem.  I believe the intention is to show that expression of RILP results in the reorganization of ORP1L.  What would be really helpful is to show that ORP1L moves from an ER compartment (first set of panels; highlighted by VAP-A colocalization, as shown) to a late endosomal/lysosomal compartment (here you need a Rab7/lysotracker marker).  It really looks like ORP1L is moving but there is only one way to prove this: show colocalization with a late endosomal/lysosomal marker!  This is where the fluorescence microscopy could be valuable.  It is also really helpful here to show statistics (100 cells counted, so many interactions without RILP, so many interactions with RILP), and calculate some kind of statistical significance.  Then, this would be really solid.  Otherwise it is speculation.  

R: Thank you for your comment.

We agree that it would be better label the late endosome/lysosome with late endolysosomal markers. Due to technological limitation, we can’t process 4 fluorophores labeling, it is hoped you will understanding this limitation. However, lots of papers have proved ORP1L locates at the late endosome/lysosome, suggesting ORP1L is an endosomal associated protein (the papers were cited in the manuscript). In addition, we showed that ORP1L normally associated with the endocytic compartments, mainly the late endodosmes/lysosomes marked by Lamp1, and we added this data in the supplemental materials as Figure s1A (also appended below). In Figure 2E, RILP induced ORP1L perinuclear clustering to inhibit ORP1L colocalizing with VAPa, as mentioned above, ORP1L associates with the late endosome/lysosome, thus we propose that RILP inhibits the endolysosome-ER contact. As most of the cells appeared the similar phenotype, so we didn’t carried out quantitative analysis.

Line 300-305: I would be very careful about speculating that knocking down Rab7 has no effect on RILP localization.  I see no data showing the degree of Rab7 knockdown (western blot), as even a small amount of Rab7 may be sufficient.  On the other hand, there may be "hidden effects" of Rab7 knockdown that we do not see, that greatly affect the cells.  It is a leap of faith to suggest that other Rabs are involved based on this thin amount of data.    

R: Thank you for your comment.

The data showing Rab7 knockout was provided in Figure s1E (also appended below, and sh-3 was used in our experiments). We agree that probably there exists hidden effects of Rab7-knockdown and other Rabs may be involved, which are far from our present story, and need comprehensive investigations.

Figure 4B: there is no Zoom here.  This would be helpful.

R: Thanks for your suggestion. We added an insert for zoom area.

 I renew my objection to the filipin staining.  I gave the authors examples of what filipin staining looks like.  Also, there is no demonstration of how often this interaction occurs, or whether the authors have chosen the one picture that looks good.  There is no details of how many cells were counted here or of any statistics.   Furthermore, filipin stains free cholesterol.  That means, the authors are showing a perinuclear clustering of late endosomes/lysosomes with an accumulation of free cholesterol.  This is a lysosomal storage condition.  

R: Thanks for your comment.

Filipin staining was carried out according to the protocol from the previous described (paper cited in the text). As most of the cells exhibit the similar phenotype, we think it is not necessary to conduct quantitative analysis. Yes, Filipin stains free cholesterol (we indicated this description in the text in the revised manuscript), our observation demonstrated that RILP induces perinuclear clustering of late endosomes/lysosomes with an accumulation of free cholesterol, which means RILP inhibits cholesterol transport.

Figure 6: Although I still don't understand how late endosomes/lysosomes undergo autophagy, and then what happens to the free cholesterol. I accept that it appears to be true under the conditions and accept that the authors prefer to keep this figure.  Figure 6M is so small to be almost meaningless.  

R: Thanks for your comment.

As suggested, We have modified Figure 6M.

Figure 7: Although I still have some misgivings about the rationale for the choice to use ORP7, I accept the authors decision to keep this figure.   I would ask the authors to deliberately distinguish between total cholesterol, free cholesterol, and cholesteryl ester, the many times that it is mentioned in the text.  I still think the authors use the term cholesterol too loosely, as if they are all the same.    

R: Thanks for your comment.

Filipin stains free cholesterol, we added this description in the main text.

The supplemental files were in a .rar format and I could not access them. 

R: As the supplemental materials contains several files of different types, and only one file can be uploaded in the submission system, therefore we uploaded the files in a Rar fold, you can download the files from the system. Also, in this response, you can find the added figure s1A. other supplemental figures were not modified.

Comments on the Quality of English Language

Just some minor errors.  For example: last line of abstract

"Our results suggest that RILP interferes ER-endolysosome interaction " should be

"Our results suggest that RILP interferes with ER-endolysosome interaction..."

R: Thanks for pointing out this error. We have corrected this error.
